# A General Framework to Boost 3D GS Initialization for Text-to-3D Generation by Lexical Richness

Lutao Jiang[*]
The Hong Kong University of Science
and Technology (Guangzhou)
Guangzhou, China
ljiang553@connect.hkust-gz.edu.cn

Hangyu Li[*]
The Hong Kong University of Science
and Technology (Guangzhou)
Guangzhou, China
hli886@connect.hkust-gz.edu.cn

Lin Wang[†]
The Hong Kong University of Science
and Technology (Guangzhou)
Guangzhou, China
The Hong Kong University of Science
and Technology
Hong Kong SAR, China
linwang@ust.hk

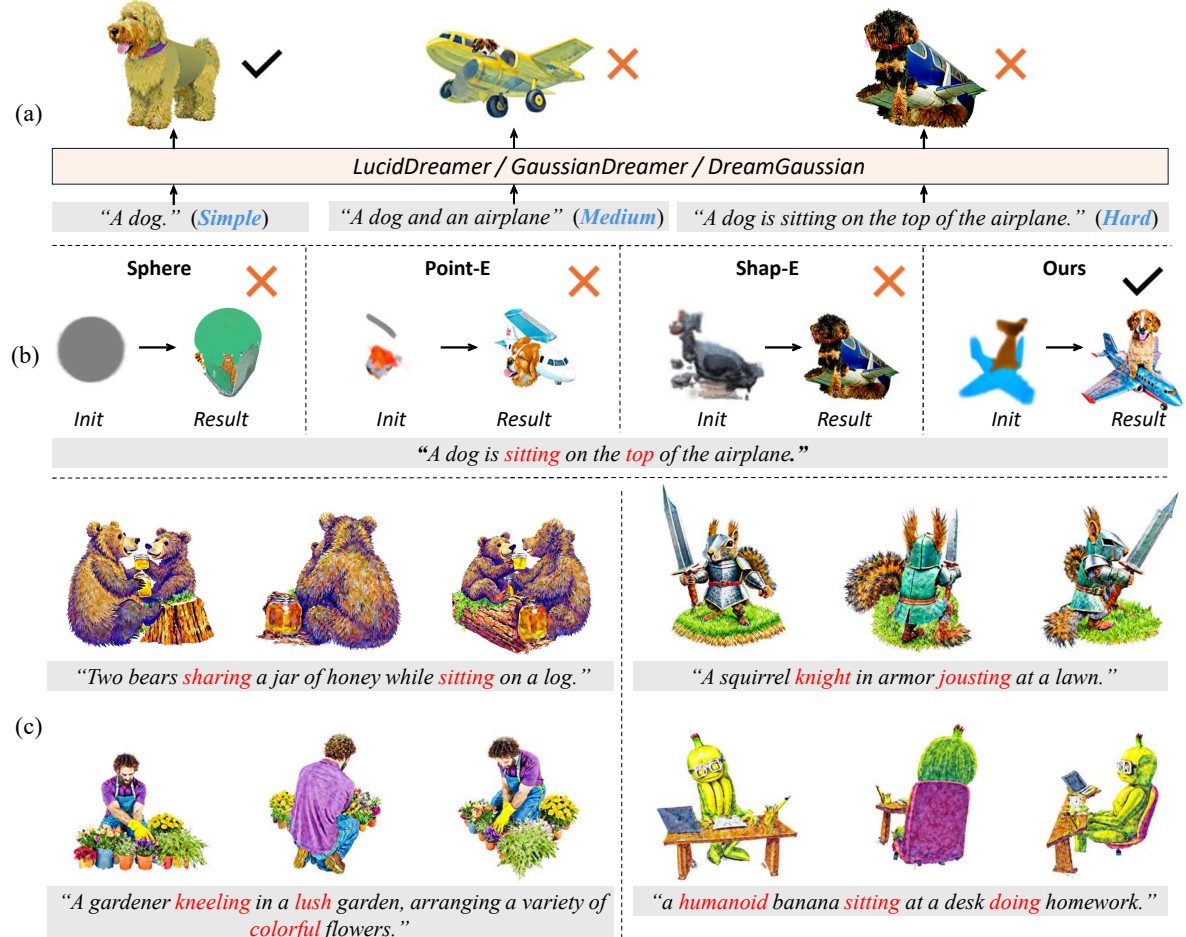

**Figure 1: (a) Comparison of rendered results with the increasing lexical richness of texts (simple, medium, hard), based on SoTA GS-based models, *e.g.*, LucidDreamer [14]. It turns out that these models fail to generate the correct shapes for the lexically complex texts. (b) Comparison of different initialization methods. (c) Rendered results of our initialization + LucidDreamer.**

[*]Equal contribution.
[†]Corresponding author.

*MM '24, October 28-November 1, 2024, Melbourne, VIC, Australia*

© 2024 Copyright held by the owner/author(s). Publication rights licensed to ACM.
ACM ISBN 979-8-4007-0686-8/24/10
https://doi.org/10.1145/3664647.3680740

## Abstract

Text-to-3D content creation has recently received much attention, especially with the prevalence of 3D Gaussians Splatting (3D GS). In general, GS-based methods comprise two key stages: initialization and rendering optimization. To achieve initialization, existing works directly apply random sphere initialization or 3D diffusion models, *e.g.*, Point-E, to derive the initial shapes. However, such strategies suffer from two critical yet challenging problems: 1) the final shapes are still similar to the initial ones even after training; 2) shapes can be produced only from simple texts, *e.g.*, '*a dog*', not for lexically richer (or harder) texts, *e.g.*, '*a dog is sitting on the top of the airplane*'. To address these problems, this paper proposes a novel general framework to boost the 3D GS Initialization for text-to-3D generation upon the lexical richness. Our key idea is to aggregate 3D Gaussians into spatially uniform voxels to represent complex shapes while enabling the spatial interaction among the 3D Gaussians and semantic interaction between Gaussians and texts. Specifically, we first construct a voxelized representation, where each voxel holds a 3D Gaussian with its position, scale, and rotation fixed while setting opacity as the sole factor to determine a position's occupancy. We then design an initialization network mainly consisting of two novel components: 1) Global Information Perception (GIP) block and 2) Gaussians-Text Fusion (GTF) block. Such a design enables each 3D Gaussian to assimilate the spatial information from other areas and semantic information from texts. Extensive experiments show the superiority of our framework of high-quality 3D GS initialization against the existing methods, *e.g.*, Shap-E, by taking lexically *simple*, *medium*, and *hard* texts. Also, our framework can be seamlessly plugged into state-of-the-art training frameworks, *e.g.*, LucidDreamer, for semantically consistent text-to-3D generation. The project code is available at https://vlislab22.github.io/DreamInit/.

## CCS Concepts

• **Information systems → Multimedia content creation**.

## Keywords

Text-to-3D Generation, Initialization of 3D Gaussians

**ACM Reference Format:**
Lutao Jiang, Hangyu Li, and Lin Wang. 2024. A General Framework to Boost 3D GS Initialization for Text-to-3D Generation by Lexical Richness. In *Proceedings of the 32nd ACM International Conference on Multimedia (MM '24), October 28-November 1, 2024, Melbourne, VIC, Australia.* ACM, New York, NY, USA, 10 pages. https://doi.org/10.1145/3664647.3680740

## 1 Introduction

3D asset creation finds its applications in the realms of multimedia, such as games and Metaverse. Text-to-3D is one of the pivotal techniques that makes it possible for casual users to create semantically consistent 3D content with text inputs. Typically, benefiting from the text-to-2D image synthesis techniques [19, 20, 29, 30] with diffusion models [33], Dreamfusion [25] proposes Score Distillation Sampling (SDS) for generating 3D assets by optimizing a Neural Radiance Field (NeRF) [22] directly from the 2D diffusion models. This catalyzes a surge in research interest in NeRF-based

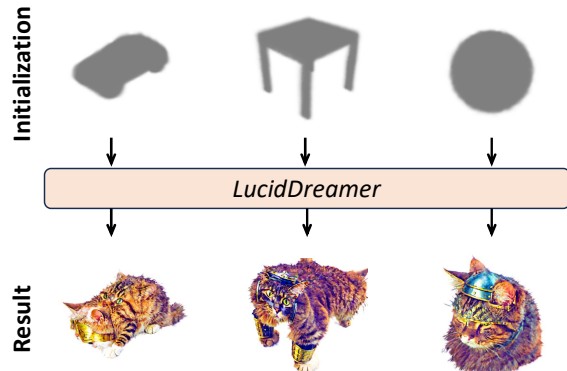

**Figure 2: Influence of different initial shapes on the training results, given the text prompt "*a DSLR photo of a cat*".**

text-to-3D generation, aiming to enhance the quality of generated content [1, 13, 15, 17, 21, 28, 37, 40, 46, 48].

Despite their success, two notable limitations persist: slow training and rendering speed. Thanks to the growing prominence of 3D Gaussian Splatting (3D GS) [12], it is dominantly adopted as the representation for faster training and rendering [3, 5, 14, 35, 41, 44]. The common paradigm of these methods consists of 1) initialization and 2) rendering optimization. The initialization stage starts from a randomly initialized sphere or generated point cloud from Shap-E [10] or Point-E [23]. In the rendering optimization stage, the SDS loss (or its variant) is used as the supervision.

While the GS-based methods demonstrate impressive generation quality, they mainly focus on the text that only contains a single entity and thus fail to generate plausible results for the lexically richer texts. Fig. 1(a) illustrates the variation in generation difficulty across different levels of lexical richness for the state-of-the-art (SoTA) GS-based models, *e.g.*, LucidDreamer [14], with the initialization methods, *e.g.*, Shap-E. To better measure the generation difficulty, we categorize text inputs into three levels based on their lexical richness: **1)** Simple: text merely contains one entity, *e.g.*, *"A dog"*. **2)** Medium: text contains multiple entities but without spatial or interaction relationships, *e.g.*, *"A dog and an airplane"*. **3)** Hard: text contains multiple entities with complex spatial or interaction relationships, *e.g.*, *"A dog is sitting on the top of the airplane"*.

Upon investigating the underlying reason for this phenomenon, we identify that 3D GS-based methods exhibit a pronounced sensitivity to shape initialization. Taking LucidDreamer as an example in Fig. 2, the optimized shapes are still similar to the initial shapes, *car*, *table*, and *sphere*. As a result, given that Point-E and Shap-E are limited to generating objects for simple text, the existing methods relying on their initialization face challenges in generating a semantically consistent 3D shape. The reason is that the absence of a lexical-rich dataset for training Point-E and Shap-E results in their inability to generate initial shapes that accurately reflect texts with complex lexical structures. As shown in Fig. 1(b), this restriction becomes the critical bottleneck and hinders these GS-based models' ability to accurately generate complex shapes.

In light of this, this paper explores a crucial yet challenging question: how to develop a general framework to create high-quality initial 3D shapes for semantically consistent text-to-3D upon the

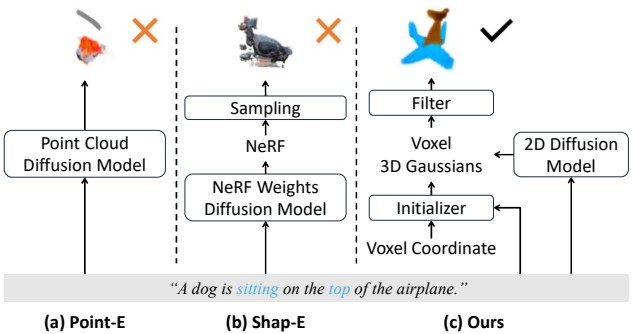

**Figure 3: Comparison of different initialization methods.**

lexical richness. Fig. 1(b) shows that taking our initialization, LucidDreamer can generate semantically consistent 3D content even with *hard* text inputs. Moreover, as depicted in Fig. 1(c), it is also evident that our framework can generate high-quality 3D assets. To make this possible, we employ the 2D diffusion model to initialize a 3D shape, given the richer 2D text-image paired data compared to 3D data. Based on this, our key idea is *to aggregate 3D Gaussians into spatially uniform voxels to represent complex shapes while enabling the spatial interaction among the 3D Gaussians and semantic interaction between Gaussians and texts*. Specifically, we first construct a voxelized representation, where each voxel contains a Gaussian with the position, scale, and rotation fixed. Only the opacity and color can be updated during training. Thus, the shape is determined solely following the opacity, which reduces the complexity significantly. This spatial-uniform representation also enables it to represent the more complex scenes (Sec. 3.2.1). As for information interaction, we design a novel initialization network consisting of two core blocks: the Global Information Perception (GIP) block and the Gaussians-Text Fusion (GTF) block. The purpose of the GIP block is to construct a spatially global information interaction among the voxelized 3D Gaussians (Sec. 3.2.2). However, establishing pair-wise interactions among each Gaussian presents a computational challenge of $O(n^2)$, where $n$ denotes the total number of 3D Gaussians. Therefore, considering the interaction among coarse areas is enough, we equally partition the space into $16^3$ 3D grids for simplification and implement a self-attention mechanism among these subdivisions. The GTF block aims at enhancing the semantic consistency of lexical-rich texts by fusing the Gaussian feature and text feature (Sec. 3.2.3). By utilizing a cross-attention mechanism, this block can effectively bind each 3D Gaussian to a more related word for better feature fusion. A comparison of different initialization methods is shown in Fig. 3.

To demonstrate the effectiveness of our framework, we set three groups of comparisons: **1)** Comparisons of different initialization methods. **2)** Comparisons of rendered results using the same SoTA GS-based model, *e.g.*, LucidDreamer with different initialization methods; **3)** Comparisons of the rendered results among different SoTA GS-based models (*e.g.*, LucidDreamer, DreamGaussion and GaussianDreamer) based on our initialization. All these experimental results demonstrate the superiority of our general framework.

In summary, our contributions are three-fold: (**I**) We comprehensively analyze the importance of the initialization of the SoTA GS-based text-to-3D methods and examine their generation capacities

upon the lexical richness of input texts. (**II**) We propose a general framework for lexically richer text-to-3D generation. The proposed initialization framework can be plugged into other GS-based text-to-3D methods. (**III**) We propose the voxelized 3D Gaussians for better initialization. Moreover, we design the Global Information Perception block and the Gaussians-Text Fusion block to impose the spatial interactions among 3D Gaussians and semantic interactions between Gaussians and texts. (**IV**) Extensive experiments prove that our method can provide a feasible initial shape for subsequent training of SoTA text-to-3D GS methods.

## 2  Related Work

**3D Gaussians Initialization.** The traditional 3D Gaussian Splatting method utilizes the SfM algorithm [31] to get the initial sparse key points from multi-view images. However, the text-to-3D generation can't provide these. Therefore, for GS-based text-to-3D, there are three mainstream initialization methods in total, random sphere, Point-E [23], Shap-E [10]. DreamGaussian [35] applies a random sphere initialization, thereby limiting their capacity to generate the complex shape. GaussianDreamer [44] designs an algorithm to employ Shap-E to obtain the initial point cloud. LucidDreamer [14] directly utilizes a Point-E to generate point cloud as its initialization. However, all these methods fail to deal with lexical-richer texts. On the contrary, our method squeezes the ability of the 2D diffusion model for 3D initialization, significantly promoting the lexical richer initialization.

**Text-to-3D based on 3D GS.** Recently, the appearance of differential 3D representation methods such as NeRF [22], DMTet [32], 3D Gaussian Splatting [12], *etc*, construct the foundation for text-to-3D. SDS loss [25] provides the key technique to optimize these differential representations from a 2D diffusion model given text input. And extensive researches follow it to improve based on NeRF [2, 4, 7, 8, 16–18, 26, 38, 39, 42, 47]. Due to the extremely high rendering speed and training speed of 3D GS representation, an increasing number of text-to-3D frameworks [5, 6, 9, 14, 35, 43–45] apply the 3D GS representation as their 3D model. DreamGaussian [35] proposes a two-stage training method consisting of 3D GS training and mesh appearance finetuning extracted by 3D GS. However, the poor performance of the geometry damages the mesh appearance finetuning. GSGEN [5] utilizes 2D SDS and 3D SDS at the same time in its method. Nonetheless, due to the lack of lexical-rich generation ability of the 3D diffusion model, the performance is relatively poor for the *hard* texts. LucidDreaer [14] proposes a novel loss called Interval Score Matching (ISM) to solve over-smoothing and low quality in 3D generation. GaussianDreamer [44] employs noisy point growing and color perturbation during initialization to further enrich the content. In this paper, we deeply focus on the initialization, which is different from the others.

## 3  Method

In Sec. 3.1, we briefly revisit the general GS-based text-to-3D pipeline. And we focus on its initialization phase. In Sec. 3.2, we introduce our proposed general framework to boost 3D GS initialization.

### 3.1  Revisiting General Text-to-3D GS Pipeline

The general training pipeline of GS-based text-to-3D methods mainly consists of: *Initialization* and *Rendering Optimization*.

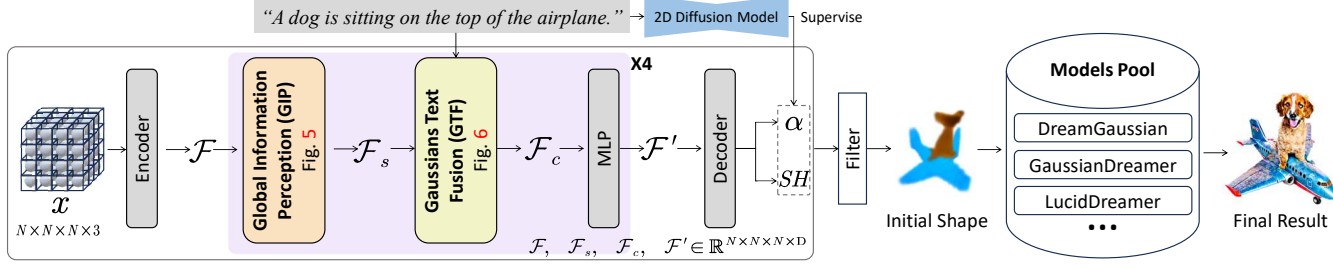

**Figure 4: Overview of the proposed framework for initialization and rendering optimization. In Phase I, an initialization network is designed to generate the initial GS shape. In Phase II, the initial GS shape can be freely plugged into one of the SoTA GS-based models (selected from the pool) for rendering optimization. "X4" means the four stacks of these modules.**

**Initialization** There are two mainstream methods for obtaining initialized shapes. The first involves randomly sampling some points to form a spherical shape. The second utilizes a 3D diffusion model, e.g., Point-E [23] or Shap-E [10], to generate initial points of a shape. Point-E is a two-stage framework. It first utilizes the text-to-image diffusion model to generate a corresponding image. Then the image is used as the input of a point cloud diffusion model to produce the 3D RGB point cloud.

As for the denoising network, transformer [36] is chosen, with the input taking the embedding of an image encoded by CLIP [27], timestep $t$, and the noisy point cloud $x_t$ to predict the noise. Shap-E first trains an encoder to encode the latent representation – parameters of the MLP. Then, a diffusion model is trained based on the latent representation. Due to the lack of complex shapes of the dataset for training, it is difficult for existing methods to generate correct shapes given the lexically richer texts. To the best of our knowledge, *our work is the first to focus on such a problem in the entire GS-based text-to-3D pipeline.*

**Rendering Optimization.** 3D Gaussian splatting[12] is superior by high inference speed, training speed, and reconstruction quality. 3D GS represents the scene through a set of Gaussian ellipsoids. The attributes of a 3D Gaussian are composed of a center $\mathbf{x} \in \mathbb{R}^3$, a scaling factors $\mathbf{s} \in \mathbb{R}^3$, a rotation quaternion $\mathbf{q} \in \mathbb{R}^4$, a color feature $\mathbf{c} \in \mathbb{R}^3$ and an opacity value $\alpha \in \mathbb{R}$, totaling 14 attributes. Given camera pose $\pi$, 3D Gaussians can be projected to 2D Gaussians, which are then used to form the corresponding image through volumetric rendering[11].

Stable-Diffusion[29] uses text prompt as a condition for generation. Ignoring the UNet Jacobian[25], the gradient of SDS loss on $\theta$ can be formulated as:

$$\nabla_\theta \mathcal{L}_{\text{SDS}}(\theta) \approx \mathbb{E}_{t,\epsilon,\pi}[\omega(t)(\epsilon_\phi(x_t, t, e) - \epsilon)\frac{\partial r(\theta, \pi)}{\partial \theta}], \quad (1)$$

where $w(t)$ is a weighting function that depends on the timestep $t$. The noise $\epsilon$ follows the distribution $\mathcal{N}(0, \mathbf{I})$. And $\epsilon_\phi(x_t, t, e)$ is the noise predicted by the network $\epsilon_\phi$ with the given text prompt $e$. $r(\cdot)$ is a differentiable renderer, $\pi$ is a camera pose, and $\theta$ represents the parameters of the renderer.

### 3.2 The Proposed Framework

Our general framework consists of two phases, including **Phase I** for 3D GS initialization and **Phase II** for rendering optimization by plugging the initialized results into one of the SoTA GS-based

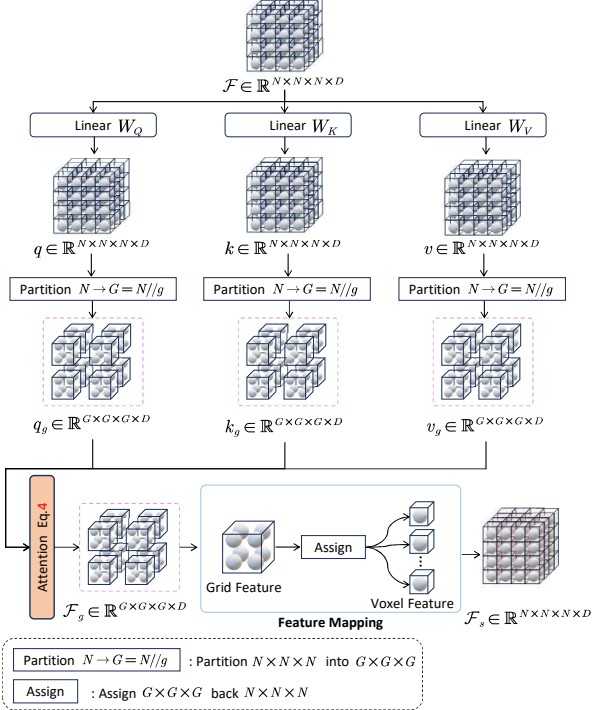

**Figure 5: The detailed illustration GIP block.**

models pool for training. Our key idea is to aggregate 3D Gaussians into spatially uniform voxels to represent complex shapes while enabling the spatial interaction among the 3D Gaussians and semantic interaction between Gaussians and texts.

**Overview.** As depicted in Fig. 4, the input of our pipeline is the given text. We first place the designed voxelized 3D Gaussians in the space (Sec. 3.2.1). Then, the coordinates $x \in \mathbb{R}^{N \times N \times N \times 3}$ of the 3D Gaussians are encoded to a higher dimensional feature $\mathcal{F} \in \mathbb{R}^{N \times N \times N \times D}$ for later interaction. $\mathcal{F}$ is then passed to our feature interaction network, which mainly consists of the Global Information Perception blocks (Sec. 3.2.2) and Gaussians-Text Fusion blocks (Sec. 3.2.3), to obtain the feature $\mathcal{F}' \in \mathbb{R}^{N \times N \times N \times D}$. Finally, we decode the feature $\mathcal{F}'$ to opacity $\alpha \in \mathbb{R}^{N \times N \times N \times 1}$ and SH color $c \in \mathbb{R}^{N \times N \times N \times 3}$. Considering that the opacity represents shape and the SH coefficient represents color, we use two separate

modules to output them.

$$\alpha = l_2^s(Softplus(l_1^s(\mathcal{F}'))), \tag{2}$$

$$SH = l_2^c(Softplus(l_1^c(\mathcal{F}'))), \tag{3}$$

where $l_{(\cdot)}^s$ means linear layer in the shape decoder, and $l_{(\cdot)}^c$ means linear layer in the color decoder. As for other properties, scale, and rotation, we set them as fixed values for faster convergence. The whole network is optimized by utilizing the 2D diffusion model [25] trained on the lexical-rich dataset. After $1K$ iterations, we filter out the positions whose opacity is less than the threshold $\tau$ and use the remaining part to formulate the initial shape. We can select one of the SoTA GS-based methods for rendering optimization in Phase II to obtain the final results. We now articulate the technical details.

*3.2.1 Voxelized 3D Gaussians Representaion.* As stated in the previous, the initial shape is a vital factor for the shape of 3D Gaussians. The final shape after rendering optimization is similar to its initial shape, which suggests that the traditional 3D Gaussians and point cloud representation are not appropriate for our target. A spatial-uniform representation can accommodate more possibilities for complex shapes. It enables to form an object at any position rather than the specific parts.

To obtain the initial shape, there is no need to construct a detailed representation. The semantic consistency is a critical factor for the initialization phase. Upon the determination of the rough shape, it is easy for the existing SoTA methods to enrich its details. Therefore, we can place the 3D voxels in space, where each voxel contains a 3D Gaussian sphere with position, scale, and rotation fixed. In this manner, the visibility of a voxel is only determined by the opacity, which significantly decreases the difficulty of training. After several iterations, we can remove the invisible 3D Gaussians, *i.e.*, whose opacity is less than $\tau$. Finally, we only record the coordinates of the remaining voxels with their colors to formulate the initial shape.

*3.2.2 Global Information Perception (GIP) Block.* This block aims to *construct the global information perception among different regions in the space.* The input is the feature $\mathcal{F}$ of each 3D Gaussian. And the output is the feature after spatial interaction among 3D Gaussians. The details of GIP block are depicted in Fig. 5.

For the target, we need to construct the global information interaction. There are two mainstream methods to achieve this, self-attention mechanism [36] or Graph Neural Network (GNN). Since self-attention can be viewed as one type of the GNNs, we choose the self-attention mechanism. However, in conventional self-attention, each 3D Gaussian needs to be scored once with all the others. If there is a total of $n$ 3D Gaussians, it needs to be calculated $n^2$ times, which results in the complexity of $O(n^2)$. Necessarily, there are tens of thousands of 3D Gaussians to form our voxelized representation, making it unacceptable. In addition, there is no necessity for such fine-detail information interaction. The grid-wise interaction can also fulfill our target. Therefore, we can simplify the point-wise self-attention to grid-wise self-attention.

Specifically, as shown in Fig. 5, we first transform the feature $\mathcal{F}$ to query $q \in \mathbb{R}^{N \times N \times N \times D}$, key $k \in \mathbb{R}^{N \times N \times N \times D}$, and value $v \in \mathbb{R}^{N \times N \times N \times D}$ through the linear transformation matrices $W_Q$, $W_K$, and $W_V$, respectively. Then, we partition the voxel space into $G^3$ grids equally. For each grid, we use the average feature vector

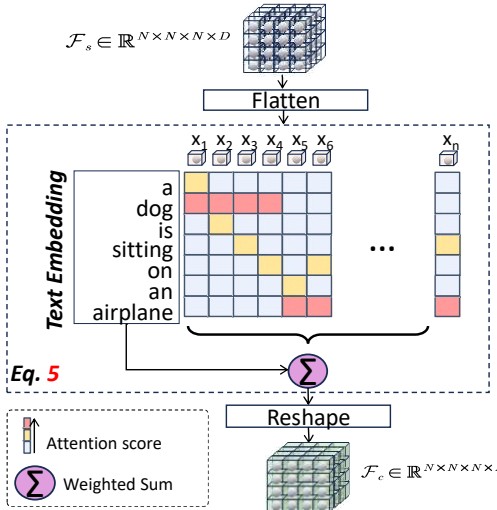

**Figure 6: The detailed illustration of GTF block.**

$q_g \in \mathbb{R}^{G \times G \times G \times D}$, $k_g \in \mathbb{R}^{G \times G \times G \times D}$, and $v_g \in \mathbb{R}^{G \times G \times G \times D}$ of the 3D Gaussians, whose center is in this grid, to represent the query, key, and value of this grid. In addition, when splitting the space, we also record the *grid_indices*, which is the index that each 3D Gaussian belongs to. We use the following equation to calculate the grid-wise self-attention [36] feature $\mathcal{F}_g \in \mathbb{R}^{G \times G \times G \times D}$

$$\mathcal{F}_g = softmax(\frac{q_g k_g}{\sqrt{D}})v_g. \tag{4}$$

Finally, grid feature $\mathcal{F}_g$ is mapped back to each 3D Gaussian feature $\mathcal{F}_s \in \mathbb{R}^{N \times N \times N \times D}$ according to the recorded *grid_indices*.

*3.2.3 Gaussians-Text Fusion Block.* The goal is to *fuse the 3D Gaussians and text features, thereby constructing the information interaction between them.* Naturally, to achieve this target, a directive way is to calculate the cross-attention by setting the Gaussian features as the query and setting the text embeddings as the key and value. The process is depicted in Fig. 6. The inputs are the text embedding and the features $\mathcal{F}_s$ of each 3D Gaussian. And the output is the fused features $\mathcal{F}_c$. Considering that the complexity of cross-attention is linear, we don't split the space into the grid to simplify the calculation.

Specifically, we first flatten the feature $\mathcal{F}_s$ to $\{x_1, x_2, ..., x_n\}$, which is the output of the last block, to the one-dimensional shape and as the query. Then, we set the text embedding $y$ to the key and value. the cross-attention is calculated by

$$\mathcal{F}_c = softmax(\frac{W_Q(\mathcal{F})W_K(y)}{\sqrt{D}})W_V(y), \tag{5}$$

where $W_Q$, $W_K$, $W_V$ are the transformation matrix of query, key, and value. In this process, the Gaussians can be assigned higher scores to their most related parts. For example, the part of the 3D Gaussians that should be composed of "*dog*" will be assigned a higher score to this word, thereby assimilating more information about this word by the weighted sum. This mechanism enables the 3D Gaussians that compose the same entity to have higher similarity, thereby achieving our target.

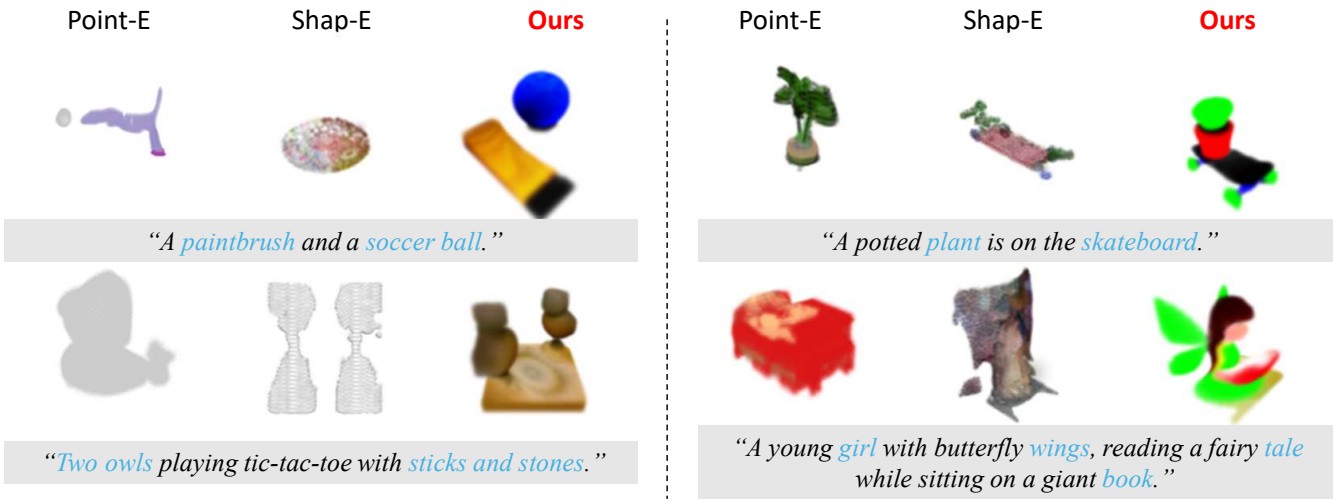

Figure 7: Comparison of different initialization methods (Point-E, Shap-E, Ours).

Table 1: The definition of the lexical richness criteria, including "simple", 'medium" and "hard".

| Lexical Richness | Definition | Example |
|---|---|---|
| "**Simple**" | The text contains just one entity. | "A dog" |
| "**Medium**" | The text contains multiple entities but without spatial relationship and interaction relationship. | "A dog and an airplane" |
| "**Hard**" | The text contains multiple entities that have complex spatial and interaction relationships | "A dog is sitting on the top of the airplane" |

## 4 Experiments

### 4.1 Implementation Details

All our experiments are conducted on a single Nvidia A40 48GB or a single RTX 3090 24GB. Our code is constructed based on PyTorch framework [24]. For the initialization phase, we use DeepFloydIF [34] to calculate the SDS loss, due to its stronger semantic understanding ability. But it can provide the supervision of just $64 \times 64$ resolution. Therefore, for the next training of DreamGaussian [35], GaussianDreamer [44], and LucidDreamer [14], we still keep consistent to their original setting, utilizing the version V2.1 of the Stable Diffusion [29], which can provide the supervision of $512 \times 512$ resolution. We set scale to $(0.05, 0.05, 0.05)$ and set rotation to $(1, 0, 0, 0)$.

### 4.2 Comparison Results

In Tab. 1, we provide the difficulty classification criteria of text. As shown in Fig. 1 (a), following the difficulty increasing, the existing methods can't generate the corresponding 3D assets. Therefore, in the following comparison, we mainly focus on the *hard* texts. **Firstly**, we compare our initialization method with the mainstream initialization methods, Point-E [23], and Shap-E [10]. **Secondly**, we compare the rendering optimization results from different initialization methods based on the LucidDreamer [14] model. **Thirdly**, we compare the different rendering optimization models based on our initialization methods, including DreamGaussian [35], GaussianDreamer [44], and LucidDreamer [14]. Finally, we provide user

studies about different initialization methods and different rendering optimization methods.

*4.2.1 Comparisons among Initialization Methods.* To demonstrate the superiority of our initialization method, we show the visual comparison in Fig. 7. The first demo is for *medium* text, and the others are for *hard* texts. It's obvious that both Point-E and Shap-E can't generate semantically consistent initial shapes for these two types of lexical richness. By contrast, our method can provide a better initialization for lexical-rich texts, which proves our effectiveness.

*4.2.2 Comparisons among Initialization Methods Based on Lucid-Dreamer.* In Fig. 8, we show the visual results of *hard* texts based on the LucidDreamer [14] model. For LucidDreamer with sphere random initialization, all test texts don't get rid of their basic shapes, which makes it difficult to generate the semantic-correct 3D assets. Moreover, it sometimes generates the content at the surface of the sphere, as shown in the demo of *kangaroos*. For Point-E initialization, though some entities exist in the results, they still lack some entities, and the relationships between entities are mistakes. While Shap-E is better than Point-E, it is also difficult to achieve the target of generating semantically consistent 3D assets. On the contrary, our initialization method demonstrates strong and robust results, which proves that our method can facilitate the development of lexical-richer text-to-3D.

*4.2.3 Comparisons among Rendering Optimization Models Based on Our Initialization.* In Fig. 9, we show the visual results of *hard* texts based on our initialization framework with DreamGaussian [35], GaussianDreamer [44], and LucidDreamer [14]. The experiments show that our method has strong generality. Our initialization method can perfectly adapt to different rendering optimization models. With our good initialized shape, we conclude that LucidDreamer and GaussianDreamer both achieve SoTA performance.

*4.2.4 User Studies.* We conduct the user studies based on a survey involving 50 participants, and 20 prompts with their corresponding 3D scene options. These scenes are generated using four initialization methods based on LucidDreamer [14], including Sphere,

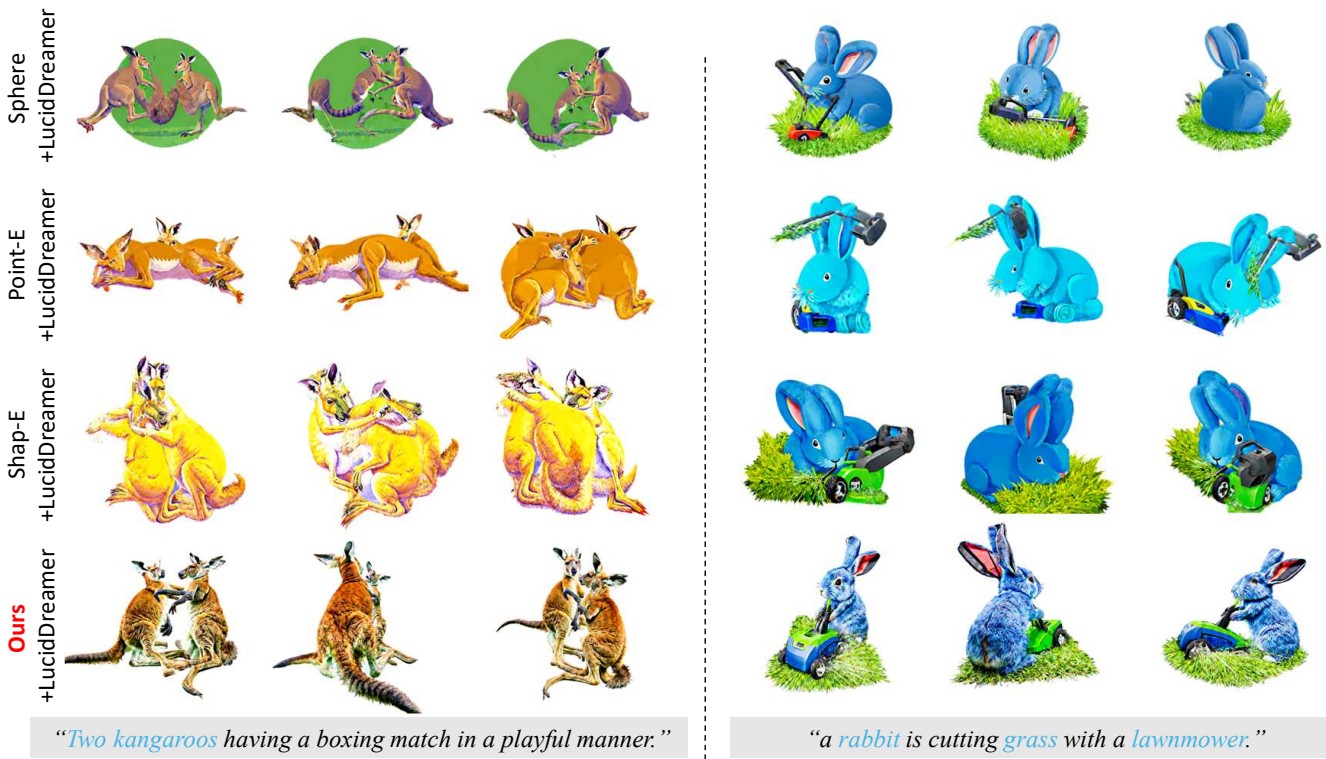

Sphere +LucidDreamer

Point-E +LucidDreamer

Shap-E +LucidDreamer

**Ours** +LucidDreamer

*"Two kangaroos having a boxing match in a playful manner."*          *"a rabbit is cutting grass with a lawnmower."*

**Figure 8: Comparison of rendered results using different initialization based on LucidDreamer [14].**

**Table 2: User preference for different initialization methods based on LucidDreamer [14].**

| Initialization | Preference |
|----------------|-----------|
| Sphere | 3.2% |
| Point-E [23] | 4.0% |
| Shap-E [10] | 4.8% |
| Ours | 88.0% |

**Table 3: User preference for different rendering optimization methods based on our initialization method.**

| Optimization | Preference |
|--------------|-----------|
| DreamGaussian [35] | 2.8% |
| GaussianDreamer [44] | 31.6% |
| LucidDreamer [14] | 65.6% |

Point-E [23], Shap-E [10], and Ours. As shown in Tab. 2, most participants (88%) think our initialization methods can provide the best generation results. Furthermore, we also compare the different rendering optimization methods based on our initialization method. As demonstrated in Tab. 3, the users preferred our initialization with LucidDreamer.

## 4.3 Ablation Study

*4.3.1 The Training Process of Initialization.* In Fig. 10, we provide the detailed process of our initialization training. With only 1, 000 iterations (12 minutes and 20 seconds), our method can generate the semantically consistent initialization shape.

*4.3.2 The Validation of Blocks.* To validate the effectiveness of our designed GIP block and GTF block, we conduct the ablation studies. As demonstrated in Fig. 11 (a), we show the comparison of *complete design*, *w/o GTF*, and *w/o GIP*. It is obvious that without the GIP block, the man is standing in a spare area, which loses some semantic consistency. A similar tendency is shown without the GTF

block. Our complete design can enable the initialized shape to have a semantic consistency with the text.

*4.3.3 The Number of Grid in Grid-wise Self-attention.* As shown in Fig. 11 (b), we validate the choice of the number of grids $G$ when partitioning the space in the GIP block, according to the quality of the initialized shape. When setting $G$ to 16, the connection line between the man and the lawnmower can be partly initialized. However, when setting the $G$ to 12, it disappears. When $G$ decreases, the semantic consistency also decreases. Therefore, taking the balance between the calculation burden and the effectiveness into consideration, we set $G$ to 16.

*4.3.4 Whether to Fix the Scale and Rotation.* To decide whether to fix the attributes of scale and rotation in our initialization phase, we compare the convergence speed and the initialized quality in Fig. 11 (c). With the scale and rotation fixed, we can finish a semantically consistent initialization using only about 12 minutes. On the contrary, without the scale and rotation fixed, even though it costs 20 minutes, it still fails to generate a feasible initialized shape. Consequently, we can attribute this phenomenon to that fixing the scale and rotation can significantly reduce the training difficulty. Due to the clear objective, by only determining the occupancy of a voxel, we can finish our target rapidly.

## 5 Conclusion

In this paper, we analyzed the importance of the initialization in GS-based text-to-3D frameworks. We then proposed an initialization framework that can be viewed as a plug-and-play initializer for

Ours + DreamGaussian
Ours + GaussianDreamer
Ours + LucidDreamer

*"A stylish fox typing on a vintage typewriter."*

*"A whale breaching the ocean surface and splashing back down."*

*"a man reclining in a cozy armchair, watching his favorite movie on a tablet."*

*"A knight is setting up a campfire.."*

**Figure 9: Comparison of rendered results with different training frameworks based on our initialization.**

100 iter 1min14s    300 iter 3min42s    500 iter 6min10s    700 iter 8min28s    900 iter 11min6s    1000 iter 12min20s

**Figure 10: The initialization training process of our method. We use** *"A knight is setting up a campfire."* **for illustration.**

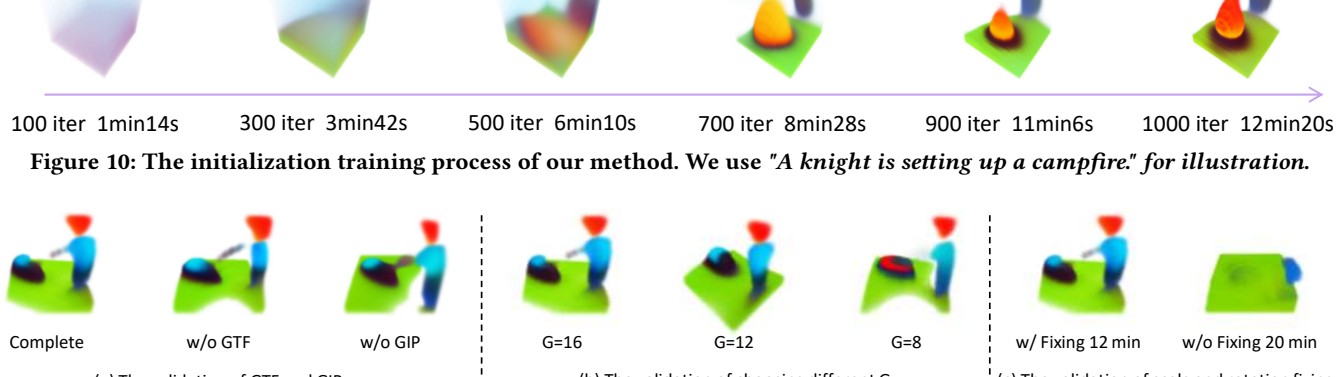

Complete    w/o GTF    w/o GIP    G=16    G=12    G=8    w/ Fixing 12 min    w/o Fixing 20 min

(a) The validation of GTF and GIP    (b) The validation of choosing different G    (c) The validation of scale and rotation fixing

**Figure 11: The ablation studies. The prompt is "a man cutting grass with a lawnmower".**

the different GS-based text-to-3D methods. To achieve the rapidly semantic-consistent initialization, we proposed the voxelized 3D Gaussians representation and designed an initialization network consisting of the GIP block and GTF block. Extensive experiments show that our method can produce the initial shape for the lexical-richer text, facilitating the development of this field significantly.

**Future Work:** Solving the multi-face Janus problem is particularly difficult when only supervised by a 2D diffusion model, especially for lexically richer (*i.e.*, *medium* and *hard*) texts. Introducing some other methods related to this is a valuable exploration direction. Future work will be devoted to solving this problem.

## Acknowledgements

This paper is supported by the Guangzhou Fundamental and Applied Basic Research Fund (Grant Number: 2024A04J4072) and HKUST (GZ) HPC Platform.

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
