# OpenReview forum: "A General Framework to Boost 3D GS Initialization for Text-to-3D Generation by Lexical Richness"
_acmmm.org/ACMMM/2024/Conference — MM2024 Poster_

### Official Review · Reviewer_ip5p · 2024-05-03

**Rating:** 4
**Confidence:** 3

**Summary:**

This paper proposes a novel general framework to boost the 3D GS Initialization for text-to-3D generation upon the lexical richness. The key idea is to aggregate 3D Gaussians into spatially uniform voxels to represent complex shapes while enabling the spatial interaction among the 3D Gaussians and semantic interaction between Gaussians and texts. The authors design an initialization network mainly consisting of two components: the Global Information Perception (GIP) block and Gaussians-Text Fusion (GTF) block. Such a design enables each 3D Gaussian to assimilate the spatial information from other areas and semantic information from texts.

**Strengths:**

This paper analyzes the importance of initialization for state-of-the-art GS-based text-to-3D methods and examines their generation capabilities based on the lexical richness of input texts. The authors propose voxelized 3D Gaussians for better initialization, which can be integrated into other GS-based text-to-3D methods. Moreover, a Global Information Perception block and a Gaussians-Text Fusion block were designed to facilitate spatial interactions among 3D Gaussians and semantic interactions between Gaussians and texts, respectively.
The experiments demonstrate the influence of initialization on the quality of 3D generation, as well as the effectiveness of the initialization method proposed in this paper.

**Limitations:**

Figure 4 lacks a detailed explanation, such as the meaning of "X4" in it.

What does the 'g' represent in Figure 5? You state that in Equation 1, 'g' denotes a differentiable renderer. The abuse of reusing symbols without providing clear definitions for their meanings in different contexts would lead to confusion among readers.

The visual quality of the generated results is not impressive, compared with other methods in Figure 8.

**Suitability:**

2

---

### Official Review · Reviewer_P4sF · 2024-05-24

**Rating:** 4
**Confidence:** 3

**Summary:**

The paper presents a novel framework aimed at enhancing the initialization process in 3D Gaussian Splatting (3D GS) for text-to-3D generation, addressing the challenges posed by lexically rich text inputs. The authors propose aggregating 3D Gaussians into spatially uniform voxels and introducing an initialization network with Global Information Perception (GIP) and Gaussians-Text Fusion (GTF) blocks. This framework ensures that even complex textual descriptions can be converted into accurate and semantically consistent 3D shapes. The paper provides extensive experiments demonstrating the framework's superiority over existing methods, such as Point-E and Shap-E, and showcases its integration into state-of-the-art models like LucidDreamer.

**Strengths:**

+ Novelty and Innovation: The introduction of voxelized 3D Gaussians and the specialized initialization network components (GIP and GTF blocks) represent a significant advancement in the field of text-to-3D generation. This approach addresses the critical issue of generating semantically accurate shapes from complex text descriptions.
+ Flexibility and Compatibility: The proposed framework is designed to be plug-and-play, allowing it to be integrated seamlessly with existing state-of-the-art models. This flexibility enhances its practical applicability and potential for widespread adoption.
+ Addressing a Critical Challenge: The focus on improving initialization for lexically richer texts fills a significant gap in the current research landscape, where existing methods struggle with complex text-to-3D tasks.

**Limitations:**

- Lack of Time Comparison: The paper does not provide a detailed comparison of the time required for the proposed initialization method versus existing methods. Including a time efficiency analysis would strengthen the argument for the practical advantages of the proposed framework.
- Lack of Quantitative Comparisons: The evaluations are largely qualitative, relying on visual comparisons. Incorporating more quantitative metrics, such as accuracy, consistency, or error rates, would provide a clearer, more objective measure of the framework's performance.
- Limited Lexical Richness Testing: While the framework shows impressive results for complex texts, the examples provided are still relatively constrained in scope. Future work could include testing with an even broader range of complex and abstract text descriptions to fully validate the method's robustness.
- Computational Complexity: The approach involves a high computational cost, particularly during the initialization phase with the voxelized representation and self-attention mechanisms. The paper could benefit from a more detailed analysis of the computational efficiency and potential optimizations.
- Dependence on 2D Diffusion Models: The reliance on 2D diffusion models for initial shape generation might limit the framework's flexibility. Exploring alternative methods for this initial step could enhance the framework's adaptability.
- Future Work Direction: The paper briefly mentions the Janus problem and its complexity when supervised by a 2D diffusion model. Providing a more detailed discussion on potential solutions or directions for future research would strengthen the conclusion and highlight the ongoing challenges in this domain.
- Insufficient Comparisons with Other Text-to-3D Methods: The paper lacks a comprehensive comparison with other state-of-the-art text-to-3D methods such as Dreamfusion, ProlificDreamer, GSGen, MVDream, and GeoDream. Including these comparisons would provide a more thorough evaluation of the proposed framework's relative performance.

**Suitability:**

3

---

### Official Review · Reviewer_EWpD · 2024-05-26

**Rating:** 5
**Confidence:** 4

**Summary:**

The paper is target at generating 3D GS that aligns with rich text input. It focuses on the initialization stage of the generation, and proposes a 3D GS initialization method which adapts to existing text-to-3DGS creation methods based on SDS. In the initialization stage, the paper models 3D Gaussian as a voxel representation, which learns the 3D spatial relationship and semantic association by doing self attention among grids, and cross attention with text embeddings, and then decode and render to initial shape. The obtained representation can be used as the initialization of existing Text-to-3DGS methods, and its spatial representation aligned with the text input can enable the final generated 3DGS to be aligned with rich semantics.

**Strengths:**

- This article proposes an initialization method rich in semantics and spatial geometric relationships for 3DGS generation, which has an auxiliary effect on the 3D generation pipeline of Score Distillation (SDS).
- The article uses self attention between grids and cross attention between grid and text embedding to effectively implement 3D geometric initialization with semantic information. (Theoretically, the semantic association can be visualized using methods like PCA, which will be a demonstration of the effectiveness of the modeling)

**Limitations:**

- The article proposes at line 109 of the abstract that existing methods have the problem that the final shapes are still similar to the initial ones even after training. However, the current framework only improves the initialization and does not seem to be able to solve this post-stage optimization problem.
- The article mentions in lines 451/452 that the initial stage is also based on the powerful image prior for per-scene optimization. So for the problem that the generated results are difficult to align with complex text input that the paper wants to solve, can text-to-image-to-3DGS solve it easily? It may be necessary to consider whether the image-to-3D method has this problem.
- Does this text-conditioned 3DGS initialization work for other applications (such as scene-level generation, feed-forward 3D generation pipeline) instead of just SDS-based 3D object generation?
- The caption of figure 6 should be corrected to GTF?

**Suitability:**

3

---

### Meta-Review · Area_Chair_2pnu · 2024-07-02

**Recommendation:** Accept (Poster)
**Confidence:** 4

**Metareview:**

This paper was reviewed by three experts in the field. The recommendations are Weak Accept, Borderline Accept, Borderline Accept. The authors have addressed most of the concerns from reviewers about the experimental comparison. Based on this, the decision is to recommend the paper for acceptance to ACM Multimedia 2024. Still, the reviewers did raise some valuable concerns. We recommend that the authors carefully read all reviewers' final feedback, and revise the manuscript as needed. We congratulate the authors on the acceptance of their paper!